# Plants Distinguish Different Photoperiods to Independently Regulate Post-Flowering Vegetative Growth and Reproductive Growth

**DOI:** 10.3390/plants14091368

**Published:** 2025-04-30

**Authors:** Weizhi Chen, Ziyi Wang, Lamei Jiang, Amanula Yimingniyazi, Cai Ren

**Affiliations:** 1College of Grassland Science, Xinjiang Agricultural University, Urumqi 830052, China; 2College of Life Sciences, Xinjiang Agricultural University, Urumqi 830052, Chinaamanula.y@xjau.edu.cn (A.Y.); 3Xinjiang Key Laboratory for Ecological Adaptation and Evolution of Extreme Environment Biology, College of Life Sciences, Xinjiang Agricultural University, Urumqi 830052, China

**Keywords:** photoperiod, post-flowering, vegetative growth, reproductive growth

## Abstract

The post-flowering stage is critical for plant yield and seed quality. This can be influenced by the photoperiod; however, the underlying mechanisms are not clear. *Arabidopsis thaliana* was selected as the experimental material to test this phenomenon. Different photoperiod treatments were implemented during the post-flowering stage to comprehensively examine the effects of photoperiod on physiological and phenotypic characteristics. This work aims to explore the photoperiod measurement mechanisms that control post-flowering growth and development. Our results showed the following: (1) During the post-flowering stage, the photoperiod had a significant impact on both vegetative and reproductive growth. (2) Photoperiod measurement mechanisms can be categorized into absolute and photosynthetic photoperiods. These mechanisms exert distinct effects. (3) Absolute photoperiod regulated the cytokinin to auxin ratio, thereby controlling the number and length of branches and the number of siliques. Extending the absolute photoperiod had a preferential promoting effect. (4) Photosynthetic photoperiod affected duration of photosynthesis. This process regulated the formation and accumulation of photosynthetic products. Consequently, it influenced the biomass and efficiency of siliques. Extending the photosynthetic photoperiod had a positive effect. This study demonstrates that plants distinguish between photoperiodic signals and energy effects to independently control post-flowering development and growth.

## 1. Introduction

Photoperiod refers to the favorable length of the day for each organism [1], and is one of the most stable ecological factors perceived by plants [2]. Since its introduction, related research has been conducted for over 100 years. Most studies of the effects of photoperiod on plant growth and development have focused on seed germination, hypocotyl elongation, and seedling growth. Significant breakthroughs have been made in understanding how photoperiod regulates flowering [3,4,5]. However, the post-flowering phase, a “critical period” due to its importance for yield and quality determination, is also regulated by photoperiod [6,7]. Furthermore, there is evidence that the flowering time and duration of the post-flowering stages may, in part, be independently controlled by photoperiod. For example, the number of days from flowering to maturity in soybeans may respond to long days even in cultivars that are insensitive to photoperiod for flowering [8,9]. Thus, clarifying the regulatory mechanism of the post-flowering stages by photoperiod is crucial for both the theoretical understanding of this process and its practical applications.

Post-flowering photoperiodic responses vary among plant species and ecotypes/genotypes [7,10,11]. For instance, in wheat studies, it has been found that the time from flowering to maturity shortens with longer light periods. This also increases the thousand-grain weight and yield, but has little impact on plant height, the number of tillers per plant, or the number of leaves per plant [12]. Similar results have also been found in other plants, such as sugar beets and triticales [13]. However, different results have been obtained for other species (e.g., soybean, rice, and potato). For example, for post-flowering photoperiod-insensitive soybeans, extended photoperiods have little effect on post-flowering growth duration [14]. However, for some photoperiod-sensitive soybeans, photoperiod can influence both vegetative growth and reproductive growth [13,15]. In some indeterminate soybean cultivars, a longer photoperiod after flowering extended the growth period and increased the number of branches. This produced more branches, flowers, and pods, resulting in higher yields [16,17,18]. Under excessively long photoperiods, in some highly sensitive varieties (soybean and adzuki bean), the whole plant may revert to vegetative growth following flower and pod abortion, and sprout new branches [19,20].

Photoperiod affects the process from flowering to maturity. However, the regulatory mechanisms are not fully understood. Studies have suggested that it is involved in the signaling and energy effects of light [21,22]. The duration of signaling and the energetic effect of light vary with photoperiod. In this case, the signaling effect refers to the time of the signaling action of light, that is, the time of the active state of the photoreceptors, which can act as a light signal even at low light intensities below the light compensation point. The energetic effect is defined as the photosynthetically active time, that is, the effective photosynthetic time above the light compensation point to fulfil the function of carbon fixation [23]. Gendron and Staiger proposed that plants sensed the photoperiod as a photoperiodic measurement mechanism and divided it into absolute photoperiod and photosynthetic period according to signaling and the energetic effect of light. At the molecular level, they verified that *Arabidopsis thaliana* employed two distinct mechanisms mentioned above to perceive photoperiod changes during seasonal flowering and vegetative growth. Specifically, the flowering pathway relied primarily on the absolute photoperiod under low light intensity to measure changes in day length. The vegetative growth pathway utilized the photosynthetic photoperiod based on the duration of effective photosynthesis to assess changes in day length [5]. In soybean research, low-light elongation and shading treatments have been applied to investigate the main mechanisms of the photoperiod response during the post-flowering stage. The direct effect involves the perception of photoperiodic signals, whereas the indirect effect is related to photosynthetically active radiation. These two effects operate independently, at least partially. For example, a direct effect significantly affected the number of nodes, seeds, and yield. The indirect effect was primarily related to the cumulative radiation from photosynthesis [18,24]. Although these terms differ from those used to describe seasonal flowering and growth, their meanings and roles in photoperiod are similar. These findings suggest that during the post-flowering stage, plants may sense photoperiods through two mechanisms: an absolute photoperiod and a photosynthetic photoperiod. However, further studies are required to confirm this hypothesis. In addition, whether these two mechanisms act independently or in coordination to regulate post-flowering growth and regulatory pathways remains an area for further research.

To explore how plants perceive photoperiod during the post-flowering growth and development stages, we carried out the following investigations on *A. thaliana* under controlled lighting conditions: (1) Does photoperiod affect post-flowering traits in plants? (2) Do plants perceive and measure photoperiod through two distinct mechanisms: absolute photoperiod and photosynthetic period? And (3) do these two mechanisms differently influence post-flowering growth and development in *plants*? If this is true, which traits are affected by each other? How do these mechanisms regulate post-flowering growth and development in *A. thaliana*?

## 2. Results

### 2.1. Photoperiod Had a Significant Effect on Post-Flowering

To investigate the role of photoperiod during the post-flowering stage, traits such as plant height, total branch number, and total branch length were measured in Experiment 1. The results showed that the photoperiod had a significant effect on many traits during the post-flowering stage. The total branch number, total branch length, silique number, and post-flowering duration increased with photoperiod extension (16NL/8D > 14NL/10D > 12NL/12D). There were no significant differences in plant height (Table 1). An extended photoperiod promoted the total branch number and total branch length, leading to an increase in the silique number owing to more flowers [17,25,26]. This process accelerated vegetative production and reproductive growth. The vegetative organ/reproductive organ biomass ratio increased with longer photoperiods. This indicated that vegetative growth was more pronounced during the prolonged post-flowering duration [27], thereby increasing plant dry weight (*p* < 0.05) (Appendix A).

### 2.2. Influence of Absolute Photoperiod on Post-Flowering

During the post-flowering stage, the absolute photoperiod significantly affected several traits, including total branch number, total branch length, silique number, silique dry weight, rate of effective siliques, seed 100-grain weight, and plant dry weight. A moderate extension of the absolute photoperiod promoted the above traits ((12NL + 2LL)/10D > 12NL/12D, (14NL + 2LL)/8D > 14NL/10D)) (*p* < 0.05), but had no significant effect on seed number per silique and plant height (Table 2 and Appendix A). The results showed that the extension of the absolute photoperiod increased the total branch number and total branch length. This led to an increase in the number of flowers, thereby forming more siliques. This extension also increased the number of effective siliques, 100-seed weight, silique dry weight, and plant dry weight.

Compared with (12NL + 2LL)/10D, the longer absolute photoperiod of (12NL + 4LL)/8D resulted in more branches and higher 100-seed weight, but decreased other traits such as silique number. This could be attributed to an excessively long absolute photoperiod, which promoted a significant increase in branch numbers. However, this led to fewer siliques owing to insufficient photosynthetic products. Consequently, the proportions of effective siliques and silique dry weights were lower (Table 2).

Post-flowering duration increased significantly with an increase in the absolute photoperiod: (12NL + 4LL)/8D > (12NL + 2LL)/10D > 12NL/12D, (14NL + 2LL)/8D > 14NL/10D). The duration of the first silique maturation–total silique maturation stage (stage 2–stage 3) was significantly extended (*p* < 0.05). The durations of the other two stages showed no significant differences (Figure 1B). These results indicated that the absolute photoperiod could affect the post-flowering duration, and a longer absolute photoperiod prolonged this duration.

Absolute photoperiod had a significant effect on photosynthetic production and endogenous hormones. NSC content and NSC accumulation are relevant indicators of photosynthate; they increased with the extension of the absolute photoperiod: (12NL + 2LL)/10D > 12NL/12D and (14NL + 2LL)/8D > 14NL/10D) (Figure 2A,B). Studies on pre-flowering have indicated that prolonging low light for a shorter time could improve photosynthetic efficiency [3]. This was consistent with our results: a longer absolute photoperiod (1, 3, 5, and 7 days after flowering) increased chlorophyll A + B content and P_n_, which could reflect photosynthetic efficiency (Appendix A) [28]. Photophotoperiod can regulate the production and distribution of endogenous hormones. A prolonged photoperiod increases the CTK content and the CTK/IAA ratio, resulting in more branches [29,30,31]. This is similar to the findings of the present study, where the CTK content and CTK/IAA ratio increased with the extension of the absolute photoperiod (12NL + 4LL)/8D > 12NL/12D). The content of IAA, an antagonistic hormone to CTK, decreased (Figure 2C–E).

### 2.3. Influence of Photosynthetic Photoperiod on Post-Flowering

Prolonged photosynthetic period significantly promoted the rate of effective silique and silique dry weight ((16NL/8D > (14NL + 2LL)/8D, 14NL/10D > (12NL + 2LL)/10D) but reduced the total branch number (*p* < 0.05). There were no significant differences in plant height, total branch length, silique number, silique length, seed number per silique, 100-seed weight, or plant dry weight. Compared with the 16NL/8D and (14NL + 2LL)/8D treatments, (12NL + 4LL)/8D increased branch number but decreased other traits. This may have been because of insufficient photosynthetic production (Table 3). There was no significant difference in the effect of post-flowering duration among the photosynthetic period treatments, even for the individual stages (Figure 2C), suggesting that the photosynthetic period did not affect post-flowering duration.

The photosynthetic period also had a significant effect on photosynthetic production and endogenous hormone content. NSC content and accumulation increased with the photosynthetic period extension: 16NL/8D > (14NL + 2LL)/8D > (12NL + 4LL)/8D and 14NL/10D > (12NL + 2LL)/10D (*p* < 0.05) (Figure 3A,B). Chlorophyll A + B content and P_n_ did not significantly increase with prolonged photosynthesis (Appendix A). These results indicated that the increase in photosynthetic products owed to an increase in photosynthetic time. The CTK and CTK/IAA ratios decreased with the prolongation of the photosynthetic period, whereas IAA increased (16NL/8D > (12NL + 4LL)/8D) (Figure 3C–E).

## 3. Discussion

The post-flowering phase constitutes a critical determinant of yield and quality, with its regulation mediated by photoperiod [6,7]. In the present study, our findings demonstrate that the post-flowering photoperiod exerts significant regulatory control over both vegetative and reproductive growth in *A. thaliana*. Mechanistically, post-flowering photoperiodic regulation can be partitioned into two distinct components: absolute photoperiod and photosynthetic period, each operating through discrete pathways.

### 3.1. Absolute Photoperiod (Signaling Effect of Photoperiod)

Our results showed that the absolute photoperiod significantly affected the post-flowering stage in terms of physiological characteristics (endogenous hormone and photosynthetic efficiency) and phenotypic traits (total branch and dry weight). Endogenous hormones are a class of organic compounds that play important roles in regulating growth and development [32]. Studies have shown that increased CTK content and CTK/IAA ratios contribute to the promotion of branch numbers before flowering [33,34]. In this study, we found that CTK content and the CTK/IAA ratio increased with the prolongation of the absolute photoperiod, thus promoting branch generation and enhancing total branch numbers. NSC content and accumulation are the key indicators of photosynthate production. NSC content increased with an extended absolute photoperiod. This may be because the photosynthetic rate and efficiency were promoted by moderately prolonging the duration of low light (Appendix A), although the effective photosynthesis time remained unchanged. This resulted in an increased plant dry weight and NSC accumulation. The vegetative/plant dry weight ratio increased with prolonged absolute photoperiods. This demonstrated that a prolonged absolute photoperiod promoted vegetative growth, thereby resulting in a longer post-flowering duration [6]. Vegetative growth was characterized by a significant increase in the total branch number and length. This led to the formation of more flowers and siliques. Sufficient photosynthate production further increased the rate of effective siliques.

The effects of absolute photoperiod can be categorized into moderate and excessive extensions for mechanistic analysis. Moderate extension enhances the total branch number and length, thereby promoting silique production. In contrast, excessive extension (12NL + 4LL/8D) paradoxically suppresses reproductive organ development despite continued branching stimulation. This suppression arises from prolonged low-light periods during the extended photophase, resulting in insufficient photoassimilate availability to sustain reproductive sink strength.

To sum up, extended absolute photoperiod elevates CTK content and the CTK/IAA ratio, thereby stimulating total branch proliferation. While maintaining photosynthetically active duration, this regime enhances photosynthetic efficiency, consequently elevating photoassimilate accumulation. These synergistic effects reinforce vegetative sink strength and prolong the post-flowering duration.

### 3.2. Photosynthetic Period (Energetic Effect of Photoperiod)

The photosynthetic period also significantly affected the post-flowering stage, yet operated through distinct mechanisms compared to the absolute photoperiod. As the photosynthetic period was prolonged, CTK content and the CTK/IAA ratio declined, while increased IAA significantly reduced the total branch number (Figure 3 and Table 3). Owing to the higher NSC content and plant dry weight, NSC accumulation increased with longer photosynthetic periods. The main reason for this change was that the extended photosynthetic period promoted the production and accumulation of photosynthetic products. Although the number of branches decreased, the total branch length remained similar. This likely resulted in a constant number of nodes, thereby keeping the number of flowers and siliques unchanged. Although the number of siliques did not change significantly after a prolonged photosynthetic period, the ratio of reproductive organs to total biomass increased. This indicated that the additional photosynthetic products were primarily directed toward the reproductive organs (Appendix A), resulting in a significant increase in silique biomass and effective silique rate.

## 4. Conclusions

The photoperiodic regulation of the post-flowering phase operates through two distinct mechanisms: absolute photoperiod and photosynthetic period. Prolonged absolute photoperiod enhances CTK biosynthesis and elevates the CTK/IAA ratio, driving increased branching. Concurrently, it improves photosynthetic efficiency, thereby amplifying vegetative growth. In contrast, an extended photosynthetic period prioritizes reproductive development by prolonging photosynthetic duration (Figure 4). This study provides a foundation for the further exploration of photoperiodic measurement mechanisms during the post-flowering stage, offering potential approaches to regulate plant reproductive time, yield, and quality through photoperiod management.

## 5. Materials and Methods

### 5.1. Plant Material and Growing Conditions

The plant material used in this study was the Columbia-0 (Col-0) ecotype of *A. thaliana*, and the seeds were provided by AraShare Biotechnology Company in Fuzhou, China. Seeds were placed in a 1.5 mL centrifuge tube and washed sequentially with 75% alcohol, distilled water, and 5% NaClO on an ultraclean table (first, shaking with a vortex oscillator for 30 s, then centrifugation with a centrifuge for 15 s, and finally pipetting the supernatant with a pipette), followed by 8 min of rest, and then washing the seeds with distilled water 5–6 times to eliminate residues. After washing, the seeds were inoculated onto an MS medium (containing 4.33 g MS medium, 30 g sucrose, and 7 g phytoagar, fixed with water to 1000 mL and pH adjusted to 5.7). To synchronize seed germination, seeds were cold stratified for 3 days at 4 °C in darkness and then transferred to an incubator (25 °C, 16 h of light/8 h of darkness at a light intensity of 130 µmol m^−2^ s^−1^) for 8 days. After eight days, seedlings were transferred to a medium containing a nutrient soil/vermiculite/perlite mixture (volume ratio of 2:1:1 and fully watered) in nutrient pots (volume of 12 cm^3^), and one seedling was transferred per pot. Finally, the seedlings were placed in a plant culture chamber (temperature 25 °C ± 2 °C, 16 h of light/8 h of darkness, and a light intensity of 130 µmol m^−2^ s^−1^) and cultured until the first flower opened [34].

### 5.2. Experimental Treatments

Plants perceive changes in the external photoperiod through the absolute photoperiod measurement mechanism, which only measures the duration of the day without considering the light intensity; when sensing external photoperiod changes through the photosynthetic period measurement mechanism, only the number of hours with a light intensity higher than the compensation point of the plant is counted (Figure 5) [5]. Based on the difference between the two measurement mechanisms, a photoperiodic environment with absolute photoperiod and photosynthetic period can be constructed by using low light (setting light intensity of 25 µmol m^−2^ s^−1^, and the light compensation point of *A. thaliana* is about 40 µmol m^−2^ s^−1^) [23,35]. The full-spectrum lamp (QSP4-060T2, Zhejiang Qiu Shi Artificial Environment Co., Ltd., Hangzhou, Zhejiang, China) employed in this study features a fixed spectrum configuration. The spectra (QE Pro, Ocean Optics, Dunedin, FL, USA) were measured and the ratios of red (600–700 nm): far-red (700–800), red: blue (400–500 nm) were calculated using a spectrometer [36]. The ratio of the red/far red was 5:1, and the ratio of the blue/red was 1:1.

The following three experiments were designed in this study: (1) normal photoperiod, that is, different day length treatments; (2) absolute photoperiod (fixing the effective photosynthetic duration and regulating the diurnal duration); and (3) photosynthetic photoperiod (fixing the diurnal duration and regulating the effective photosynthetic duration). Plants were transferred to different treatments (Table 4) at the time of the first flower opening and maintained until the entire plant was ripe.

(1) Normal photoperiod (Experiment 1): three light/dark cycle treatments (12L/12D, 14L/10D, and 16L/8D) were used under normal light conditions (light intensity of 130 µmol m^−2^ s^−1^).

(2) Absolute photoperiod (Experiment 2): The effect of the absolute photoperiod at the post-flowering stage was investigated by maintaining equal photosynthetic time and changing the illumination time to low light. This experiment was divided into two subgroups (Group I and Group II). The photosynthetic time of Group I was 12 h, and the absolute photoperiods were 12, 14, and 16 h. The specific light treatments included 12 h of normal light, 12 h of normal light + 2 h of low light, and 12 h of normal light + 4 h of low light. The photosynthetic time of group II was 14 h, and the absolute photoperiod times were 14 and 16 h. Specific light treatments included 14 h of normal light and 14 h of normal light plus 2 h of low light.

(3) Photosynthetic period (Experiment 3): The effect of the photosynthetic period at the post-flowering stage was studied by maintaining an equal absolute photoperiod and changing the photosynthetic time. The experiments were divided into two subgroups (Groups I and II). Group I: The absolute photoperiod was fixed at 16 h, and the photosynthetic time was 12 h, 14 h, and 16 h, respectively. The specific light treatments included 12 h of normal light + 4 h of low light, 14 h of normal light + 2 h of low light, and 16 h of normal light. Group II: The absolute photoperiod was fixed at 14 h, and the photosynthetic times were 12 and 14 h. The specific light treatments included 12 h of normal light + 2 h of low light, and 14 h of normal light.

### 5.3. Morphological Indicators

The post-flowering growth and development stages of *A. thaliana* from the opening of the first flower (Stage 0) were divided into the stages of first silique production (Stage 1), first silique maturation (Stage 2), and total silique maturation (Stage 3) [37]. The duration of each stage of post-flowering growth and development in the different treatment groups was observed and recorded. At full maturity of the silique, all inflorescence stem branches (including rosette branches, cauline node branches, secondary and tertiary branches, and branches longer than 1 cm in length) were counted to obtain the total number of branches, and the lengths of the various levels of branches were measured and then summed to obtain the total length of the branches. Siliques (including effective and ineffective siliques) were counted and summed to obtain the number of siliques, and silique efficiency was calculated (silique efficiency = effective silique/(effective silique + ineffective silique)). The criterion for determining the effectiveness of a silique was whether it contained seeds. The lengths of the siliques were measured using a dissecting microscope (SMZ745T, Nikon Corporation, Tokyo, Japan). Mature seeds from the siliques were collected, and the 100-seed weight was determined using an electronic balance with a precision of 0.0001 g. For the determination of biomass, the roots were first washed with water, then heated at 105 °C for 30 min, dried at 70 °C until a constant weight was achieved, and finally the dry matter mass of each part was determined by using an electronic balance with a precision of 0.0001 g.

### 5.4. Chlorophyll Content

The chlorophyll content of the rosette leaves was determined using the mixed solution method (volume ratio of pentanol:acetone was 2:1) [38]. Rosette leaves were cut into thin filaments of less than 1 mm in width and weighed with an electronic scale to obtain 0.2 g. They were placed in test tubes containing 10 mL of a 2:1 mixture of acetone and anhydrous ethanol and extracted by immersion at room temperature while being kept in the dark until the leaves were completely white. The absorbances of chlorophyll a and b were measured at 663 nm and 645 nm, respectively, and the chlorophyll a and b contents were calculated using the acetone method as follows:Chlorophyll a (mg/g) = ((12.7 × A663 − 2.69 × A645) × V)/(1000 × W)Chlorophyll b (mg/g) = ((22.7 × A645 − 4.68 × A663) × V)/(1000 × W)Total Chlorophyll (mg/g) = Chlorophyll a + Chlorophyll b
where V is the volume of the extraction solution, and W is the weight of the sample.

### 5.5. Determination of Photosynthetic Gas Exchange Parameters

Photosynthetic gas exchange parameters were quantified using a CIRAS-3 portable photosynthesis system (CIRAS-3, PP Systems, Haverhill, MA, USA). During measurements, the LED-built-in light source was maintained at 1000 µmol m^−2^ s^−1^, while ambient air with a CO_2_ concentration of 400 ± 10 µmol m^−2^ s^−1^ was supplied via a buffer bottle sampling air at 3–4 m above ground level. The net photosynthetic rate (Pn) was measured on ten plants per treatment group, with 2–3 fully expanded rosette leaves selected per plant. Each leaf underwent three sequential measurements, and values were averaged to minimize intra-leaf variability.

### 5.6. Extraction and Determination of Soluble Sugar and Starch

Dried whole plants were ground to a powder, 50 mg of this sample was added into a 10 mL graduated test tube along with 4 mL of 80% ethanol and placed in a water bath at 80 °C with constant stirring for 40 min. After cooling, the mixture was centrifuged for 10 min. The supernatant was collected in a 10 mL graduated test tube, and 2 mL of 80% alcohol was added to extract the residual sugar twice, to combine the supernatant to which 10 mg of activated carbon was added followed by an 80 °C-water bath decolorization for 30 min. Then, 10 mL of distilled water was added and filtered. 1 mL of the ethanol extraction solution was added to a test tube, followed by the addition of 5 mL of anthrone reagent, boiled in a boiling water bath for 10 min, removed and cooled. The OD density of soluble sugar was measured at 625 nm.

Next, 3 mL of distilled water was added to the residue after the extraction of the soluble sugar and stirred thoroughly. Then, the mixture was placed in a boiling water bath for 15 min. After cooling, 2 mL of cold 9.2 mol L^−1^ perchloric acid was added and stirred for 15 min. Then, distilled water was added to the test tube until the volume reached 10 mL, mixed well, centrifuged for 10 min, and the supernatant was transferred into a 50 mL volumetric flask. Then, 2 mL of 4.6 mol L^−1^ perchloric acid was added to the test tube, stirred for 15 min, 10 mL of distilled water was added, mixed well, centrifuged for 10 min, and the supernatant was transferred to a volumetric flask. The test tube was then washed with distilled water once or twice, centrifuged, and the centrifuged solution was combined in a 50 mL volumetric flask and fixed with distilled water. This solution was used to determine the starch content. The starch content of the solution was determined using an anthrone reagent and the OD values were monitored at 620 nm. The starch concentration was calculated by multiplying the glucose concentration by 0.9 [39,40].

In this study, the Non-Structural Carbohydrates (NSC) content was the sum of soluble sugar content and starch content. The accumulation of NSC was estimated by multiplying the NSC concentration by the dry weight of the organ [41,42].

### 5.7. Phytohormone Content

The stems of *A. thaliana* were sampled during the period of the first silique maturation (stage 2) and all silique maturation (stage 3), wrapped in tin foil and weighed, then stored in an ultra-low temperature refrigerator (−80 °C) for use. The levels of growth hormone auxin (IAA) and cytokinin (CTK) in the stems were measured by an enzyme-linked immunosorbent assay (ELISA) using the Shanghai Xuanzecang kit and a microplate reader (Synergy H1, USA). The results were averaged by repeating the assay four times for each sample.

### 5.8. Statistical Analyses

Differences between light treatments were determined using SPSS software (version 25.0; IBM, Chicago, IL, USA) and Tukey’s post hoc test. Different letters indicate significant differences at *p* < 0.05. Graphs were plotted using OriginPro software (version 2021, OriginLab Corp., Northampton, MA, USA).

## Figures and Tables

**Figure 1 plants-14-01368-f001:**
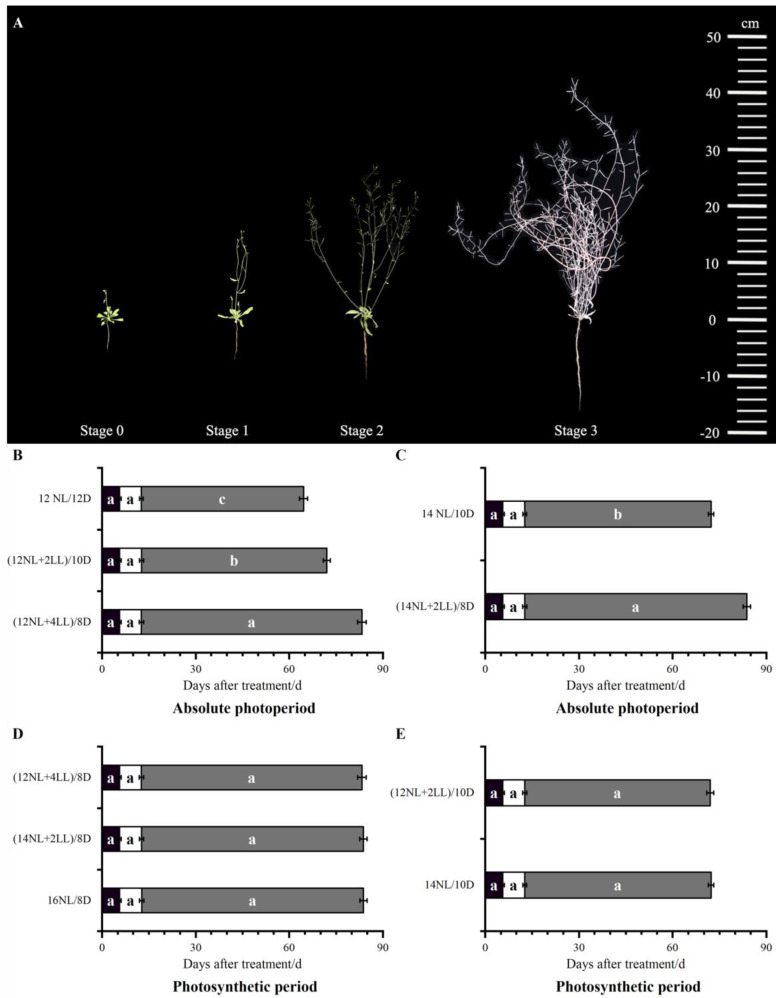
Principal post-flowering growth stage-based phenotype analysis of *Arabidopsis thaliana*: first flower opening (stage 0), first silique production (stage 1), first silique maturation (stage 2), total siliques maturation (stage 3) (**A**). The duration of reproductive phases, from first flower opening to first silique production (stage 0 to stage 1, black bars), from first silique production to first silique maturation (stage 1 to stage 2, white bars), and from first silique maturation to total silique maturation (stage 2 to stage 3, grey bars), is expressed in terms of absolute photoperiod (**B**,**C**) and photosynthetic period (**D**,**E**) during the post-flowering phase. Significant differences between treatments (*p* < 0.05) are indicated by distinct lowercase letters (one − way ANOVA with Tukey’s post hoc test). Data shown as mean ± SD (*n* = 10 plants per biological repeat, the experiment was constituted of three biological replicates).

**Figure 2 plants-14-01368-f002:**
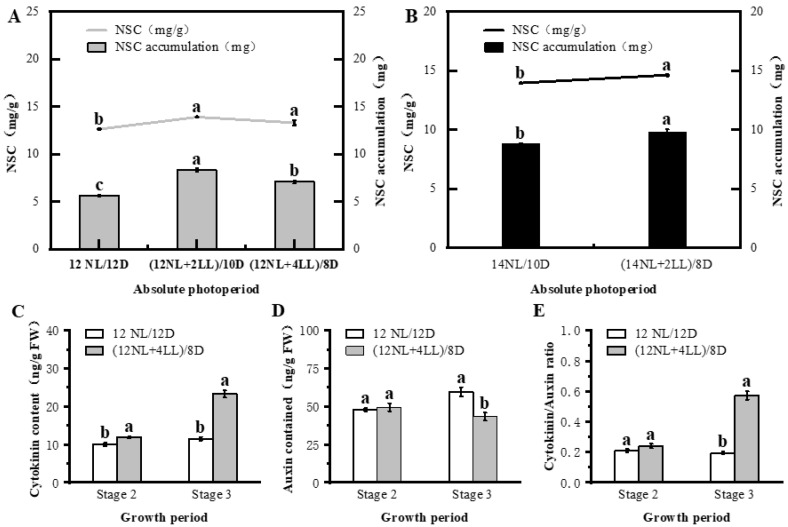
Physiological parameters as affected by different post-flowering absolute photoperiod simulations in *Arabidopsis thaliana*. Non-Structural Carbohydrates (NSC) concentration and accumulation measured in the whole plant under absolute photoperiod in group I (**A**) and II (**B**) of Experiment 2. Concentrations of Cytokinin (**C**), Auxin (**D**), and Cytokinin/Auxin ratio (**E**) in abov-ground organs at two developmental stages (stage 2, stage 3) affected by absolute photoperiod [12NL/12D, (12NL + 4LL)/8D]. Significant differences among treatments (*p* < 0.05) are indicated by distinct lowercase letters (one − way ANOVA with Tukey’s post hoc test). Data shown as mean ± SD (*n* = 4 biological replicates).

**Figure 3 plants-14-01368-f003:**
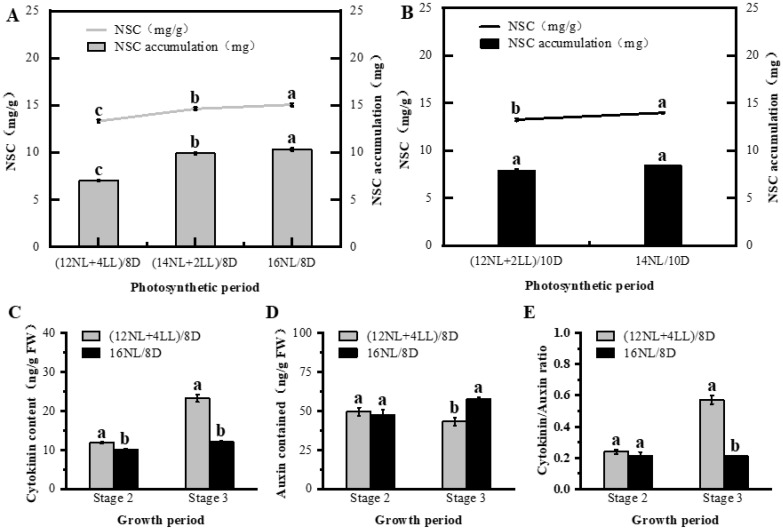
Physiological parameters as affected by different post-flowering photosynthetic photoperiod simulations in *Arabidopsis thaliana*. Non-Structural Carbohydrates (NSC) concentration and accumulation measured in above-ground organs under photosynthetic photoperiod in group I (**A**) and II (**B**) of Experiment 3. Concentrations of Cytokinin (**C**), Auxin (**D**), and Cytokinin/Auxin ratio (**E**) in above-ground organs at two developmental stages (stage 2, stage 3) under photosynthetic photoperiod ((12NL + 4LL)/8D, 16NL/8D). Significant differences among treatments (*p* < 0.05) are indicated by distinct lowercase letters (one − way ANOVA with Tukey’s post hoc test). Data shown as mean ± SD (n = 4 biological replicates).

**Figure 4 plants-14-01368-f004:**
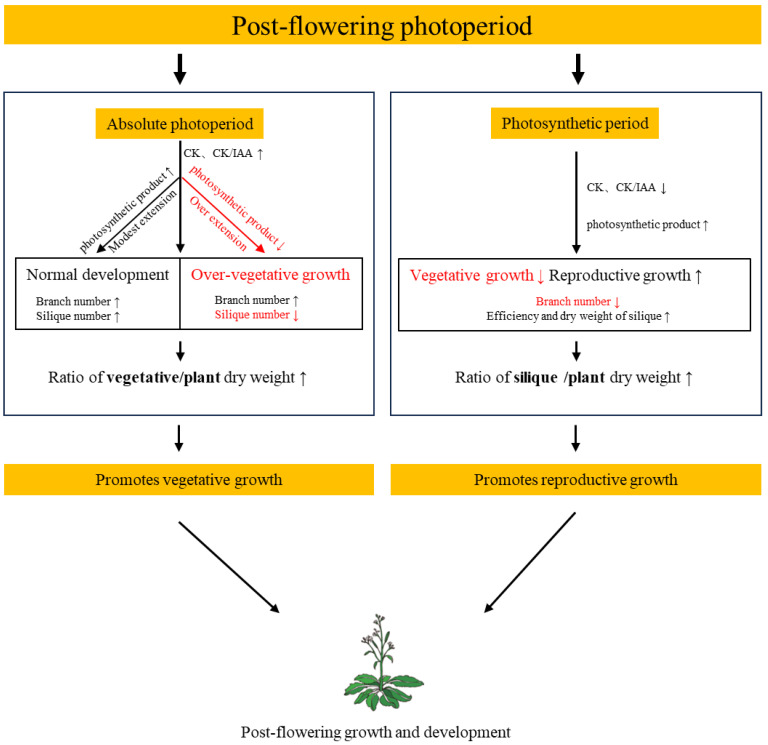
Model for two distinct photoperiod measurement mechanisms controlling post-flowering growth and development. Within a natural photoperiod, plants can distinguish absolute photoperiods from the photosynthetic period, which have distinct effects. Absolute photoperiod affects the number of branches and siliques by regulating the CTK/IAA ratio; the photosynthetic period affects silique biomass and efficiency by regulating photosynthetic product. Short black arrows indicate a positive effect, short red arrows indicate a negative effect.

**Figure 5 plants-14-01368-f005:**
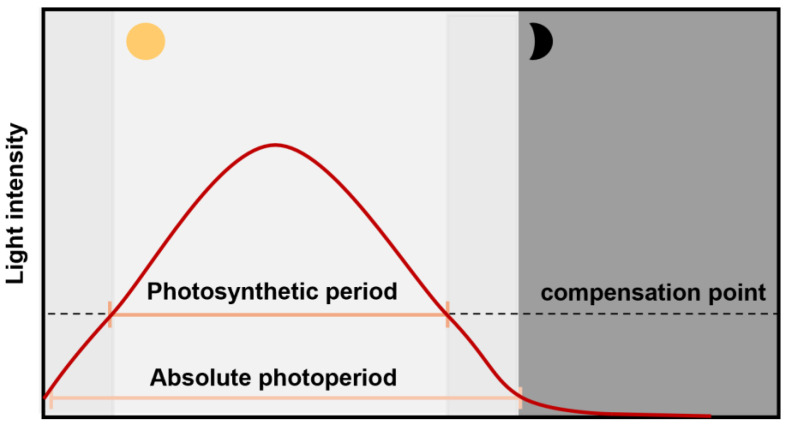
Schematic of photoperiod types measured by plants: the absolute photoperiod and the photosynthetic period [5]. Absolute photoperiod: the time of day when the photoreceptors are active, which can occur at very low light levels (lower than the compensation point) and is determined by the light sensitivity of the photoreceptors. The photosynthetic period is the time of the day when light is above the compensation point and the plant can fix carbon, which is sensed by the photosynthetic complex, and controls the metabolic daylength measurement system.

**Table 1 plants-14-01368-t001:** Morphological and developmental parameters as affected by different post-flowering photoperiod simulations in *Arabidopsis thaliana*.

Photoperiod	Morphological and Developmental Parameter				
	Plant Height (mm)	Total Branch Number	Total Branch Length (mm)	Silique Number	Post-Flowering Duration (Days)
12NL/12D	399.4 ± 15.4 a	51.2 ± 3.5 c	6325.6 ± 351.4 c	477.2 ± 25.7 c	64.6 ± 2.1 c
14NL/10D	397.0 ± 13.0 a	58.9 ± 4.3 b	6751.0 ± 467.0 b	555.4 ± 19.0 b	72.3 ± 1.7 b
16NL/8D	396.0 ± 13.8 a	62.4 ± 5.5 a	7308.0 ± 387.5 a	583.1 ± 19.7 a	83.8 ± 2.0 a

Different lowercase letters within the same column in the table indicate statistically significant differences among treatments (*p* < 0.05, one-way ANOVA, Tukey’s post hoc test). Values represent mean ± SD (*n* = 10 plants per biological repeat, the experiment was constituted of three biological replicates).

**Table 2 plants-14-01368-t002:** Morphological parameters as affected by different post-flowering absolute photoperiod simulations in *Arabidopsis thaliana*.

Group	Illumination Treatment	Morphological Parameter							
		Plant Height (mm)	Total Branches	Total Branches Length (mm)	Silique Number	Rate of Effective Silique (%)	Silique Length (mm)	Seed Number per Silique	100-Seed Weight (mg)
I	12 NL/12D	399.4 ± 15.4 a	51.2 ± 3.5 c	6325.6 ± 351.4 c	477.2 ± 25.7 c	63.7 ± 1.9 c	7.4 ± 2.1 a	18.6 ± 9.2 a	1.74 ± 011 b
	(12NL + 2LL)/10D	397.0 ± 14.2 a	62.4 ± 1.8 b	6763.8 ± 424.8 a	560.1 ± 21.4 a	69.3 ± 1.8 a	7.9 ± 2.1 a	19.5 ± 7.6 a	1.86 ± 0.05 b
	(12NL + 4LL)/8D	385.1 ± 14.5 b	67.5 ± 5.9 a	6659.3 ± 304.0 a	516.8 ± 32.9 b	67.0 ± 1.5 b	7.8 ± 2.1 a	19.1 ± 9.8 a	2.06 ± 0.11 a
II	14NL/10D	397.0 ± 13.0 a	58.9 ± 4.3 b	6750.7 ± 466.8 b	555.4 ± 19.0 b	72.1 ± 1.4 b	8.8 ± 1.9 a	23.4 ± 10.4 a	1.88 ± 0.08 b
	(14NL + 2LL)/8D	397.0 ± 12.3 a	64.5 ± 2.1 a	7292.9 ± 312.2 a	593.2 ± 21.9 a	74.4 ± 1.6 a	9.2 ± 2.1 a	25.6 ± 11.1 a	2.04 ± 0.09 a

Different lowercase letters within the same column in the table indicate statistically significant differences among treatments (*p* < 0.05, one-way ANOVA, Tukey’s post hoc test). Values represent mean ± SD (four biological replicates for 100-seed weight, three biological replicates of 10 plants each for the remaining indexes).

**Table 3 plants-14-01368-t003:** Morphological parameters as affected by different post-flowering photosynthetic period simulations in *Arabidopsis thaliana*.

Group	Illumination Treatment	Morphological Parameter							
		Plant Height (mm)	Total Branches	Total Branches Length (mm)	Silique Number	Rate of Effective Silique	Silique Length (mm)	Seed Number per Silique	100-Seed Weight (mg)
I	(12NL + 4LL)/8D	385.1 ± 14.5 b	67.5 ± 5.9 a	6659.3 ± 304.0 b	516.8 ± 32.9 b	67.0 ± 1.5 c	7.8 ± 2.1 b	19.1 ± 9.8 b	2.06 ± 0.11 a
	(14NL + 2LL)/8D	397.0 ± 12.3 a	64.5 ± 2.1 b	7292.9 ± 312.2 a	593.2 ± 21.9 a	74.4 ± 1.6 b	9.2 ± 2.1 a	25.6 ± 11.1 a	2.04 ± 0.09 a
	16NL/8D	396.0 ± 13.8 a	62.4 ± 5.5 b	7307.8 ± 387.5 a	583.1 ± 20.0 a	77.2 ± 1.3 a	9.3 ± 2.0 a	27.9 ± 13.0 a	2.06 ± 0.11 a
II	(12NL + 2LL)/10D	397.0 ± 14.2 a	62.4 ± 1.8 a	6763.8 ± 424.8 a	560.1 ± 21.4 a	69.3 ± 1.8 b	7.9 ± 2.1 a	19.5 ± 7.6 a	1.86 ± 0.05 a
	14NL/10D	397.0 ± 13.0 a	58.9 ± 4.3 b	6750.7 ± 466.8 a	555.4 ± 19.0 a	72.1 ± 1.4 a	8.8 ± 1.9 a	23.4 ± 10.4 a	1.88 ± 0.08 a

Different lowercase letters within the same column in the table indicate statistically significant differences among treatments (*p* < 0.05, one-way ANOVA, Tukey’s post hoc test). Values represent mean ± SD (four biological replicates for 100-seed weight, and three biological replicates of 10 plants each for the remaining indexes).

**Table 4 plants-14-01368-t004:** Photoperiod and lighting conditions design.

Photoperiod	Group	Light/Dark Period (h)	Illumination Treatment
Normal photoperiod (Experiment 1)	I	12L/12D	12NL/12D
		14L/10D	14NL/10D
		16L/8D	16NL/8D
Absolute photoperiod (Experiment 2)	I	12L/12D	12NL/12D
		14L/10D	(12NL + 2LL)/10D
		16L/8D	(12NL + 4LL)/8D
	II	14L/10D	14NL/10D
		16L/8D	(14NL + 2LL)/8D
Photosynthetic period (Experiment 3)	I	16L/8D	(12NL + 4LL)/8D
		16L/8D	(14NL + 2LL)/8D
		16L/8D	16NL/8D
	II	14L/10D	(12NL + 2LL)/10D
		14L/10D	14NL/10D

NL: Normal light, the light intensity is 130 µmol m^−2^ s^−1^; LL: Low light, the light intensity is 25 µmol m^−2^ s^−1^.

## Data Availability

The raw data supporting the conclusions of this article will be made available by the authors on request.

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
