# Peer review of "Plants Distinguish Different Photoperiods to Independently Regulate Post-Flowering Vegetative Growth and Reproductive Growth"

_plants, 2025, doi:10.3390/plants14091368_

Round 1
Reviewer 1 Report
Comments and Suggestions for Authors
This work explores how plants respond differentially to absolute photoperiods and photosynthetic photoperiods during post-flowering stage of growth. The authors first grow plants under normal long-day photoperiod (16h L/8h D) with normal light intensity (130 µmol m-2 s-1, typical for Arabidopsis) until plants open the first flowers. they then transfer plants to chambers conditioned with different lengths of photoperiods and/or two light intensities (normal light, and low light below photosynthetic compensation point). They closely and extensively characterize plant phenotypic traits and also determine endogenous phytohormonal levels (cytokinin and IAA) - technically tremendous work done. By integrating acquired data, they depict or correlate the response trends of plants to the two different photoperiods, which however is not at the molecular mechanism level most readers expect these days. They truly made some interesting observations, of which the most remarkable one is the significant change of cytokinin levels in response to prolonged photosynthetic photoperiod and to altered light intensities during post-flowering stage 3. Overall the work is presented in a concise and clear way.
Four major concerns.
(1) P116, the plant culture chamber has a high 25ËšC setting plus 4ËšC variation. Arabidopsis is typically grown at 22 or even 20ËšC. 25ËšC is somehow too high. Second, the 4ËšC variation is too much. It is well established that 27ËšC and above (warmer temperature) triggers plant thermomorphogenesis that elicits more stem elongation and early flowering. This temperature setting will inevitably affect growth traits and make interpretation of plants’ response to photoperiods complicated.
(2) P126, please provide the reference for claiming that ~40 µmol m-2 s-1 is the light compensation point. Is this value determined for young Arabidopsis plants during vegetative growth phase? After transitioning to reproductive growth, Arabidopsis plants may adapt to a lower light compensation point, so that even 25 µmol m-2 s-1 is above this compensation point? If so, this will challenge or void the photosynthetic period setting in this study.
(3) Table 2 shows that absolute photoperiods significantly affect total branch numbers. The 62 branches from 16L/8D vs 51 from 12L/12D treatments are a great difference. Based on authors’ definition, they count all branches that are longer than one centimeter. Arabidopsis plants grow under long-day photoperiods (16L/8D) usually develop a few primary shoots (or called rosette branches based on the authors’ definition). The transfer of plants starting flowering under LD photoperiod to the experimental setting would not increase too much primary shoot number. Then this reviewer would ask, the difference of ~10 total branch number was mainly contributed from which type of branches, cauline branch, secondary, or tertiary branch?
(4) All the photoperiod settings in this study are from neutral-day to long-day comparison (12, 14, 16h L). Why the authors did not consider, or omit, the short-day photoperiod (8h L/16h D) effect on post-flowering growth? Fig2 shows that longer photoperiods extend the stage2-to-stage3 duration. Under short-day photoperiods, does this trend hold true?
Minor points:
P367, P386 and maybe other lines, revise ‘nutritive growth’ as ‘vegetative growth’. P255, ‘nutrient organ’ changed as ‘vegetative organ’.
P107, delete ‘gun’ after pipette.
P109 -110, ‘plant agar’ should be ‘phytoagar’. The final volume should be 1000 mL, not 800 mL.
Table 1, Rows about photosynthetic period (experiment 3), column 3 about light/dark period (h), 12L/12D should be 16L/8D for “I” treatments, and 14L/10D for “II” treatments.
Fig 3C-E, the colors selected are too light and very unfriendly for readers when print in black/white mode, same for Fig 3A-B and 4A-B. Please consider using darker colors.
P325, ‘12NL/D’ revised as ‘12NL/12D’.
Author Response
“please see the attachment”

Reviewer 2 Report
Comments and Suggestions for Authors
General Comments
This manuscript presents a well-designed and clearly structured study exploring how photoperiod influences physiological and phenotypic traits in Arabidopsis thaliana. The experiments appear to be thoughtfully executed, and the overall presentation is solid. However, the discussion section lacks coherence in linking observed responses to underlying mechanisms, with many of the results presented in isolation. I strongly recommend that the authors revisit the discussion section and revise it to provide a clearer and more integrative interpretation of their findings, emphasizing cause-and-effect relationships.
Specific Comments
Introduction
Lines 82–83: There is no apparent justification for using bold font for the terms “direct effect” and “indirect effect.” Please remove the formatting to maintain consistency.
Lines 263–264: The use of the word “Repeats” is unclear in this context. Should this be “replicates” instead?
Line 285: Replace “between” with “among” to accurately reflect comparisons involving more than two groups.
Discussion
The discussion is currently too brief and lacks depth. Notably, it opens with a paragraph focusing on supplemental figures, yet the rationale behind measuring these parameters remains unclear. The authors should revisit the initial hypothesis and guide the reader through their experimental approach, clarifying how each step of the study was designed to test specific aspects of the hypothesis. This will strengthen the logical flow and enhance the interpretability of the results.
Conclusion
The conclusion section is overly long, dense, and difficult to follow. Consider condensing this section by distilling the key takeaways and focusing on the main findings that directly support the original hypothesis. A more concise and focused conclusion will greatly improve the overall clarity and impact of the manuscript.
Author Response
“please see the attachment”

Reviewer 3 Report
Comments and Suggestions for Authors
Review of the article by Weizhi Chen et al: "Plants Distinguish Different Photoperiods to Independently Regulate Post-Flowering Vegetable Growth and Reproductive Growth"
The work was carried out with Arabidopsis thaliana plants. Various photoperiodic treatments were performed after the plants bloomed. The aim of the work is to study the mechanisms of photoperiod influence that control the growth and development of plants after flowering. The results showed that: (1) The photoperiod has a significant effect on both vegetative and re-productive growth. (2) The photoperiod can be divided into categories: absolute and photosynthetic photoperiods, which have different effects on plants (3) the absolute photoperiod regulates the ratio of cytokinins and auxins, and its increase has a beneficial stimulating effect (4) The "photosynthetic" photoperiod affects the duration of photosynthesis and regulates the formation and accumulation of photosynthetic products, biomass and productivity. In this way, plants distinguish between photoperiod signals and energy effects, which allows them to independently control development and growth after flowering.
Recommendations.
1) Since the effect of low-energy light (below the light compensation point) is manifested through the effect of photoreceptors, phytochromes and cryptochromes, on plants, it is necessary to present the radiation spectrum of plants in the article and evaluate the ratio of the red/far red range of the spectrum, as well as the blue/red range. This approach will allow us to talk not only about the effect of low light intensities on plants, but also about the effect of photoreceptors on physiological and morphological parameters. 2) The phrase is not quite clear: For the determination, the light intensity of 199 the LED built-in light source was 1000 μmol m-2 s-1, and the buffer bottle was taken with 200 relatively stable air at a height of 3-4 m at a concentration of about 400 ± 10 μmol mol-1, 201 and the net photosynthetic rate (Pn) was recorded. If the rate of photosynthesis was determined at a light intensity of 1000 mmol/m2c. how can this approach be explained? Because the saturation of the photosynthetic light curve in arabidopsis occurs at lower light values.
The article is interesting, well-written, well-discussed, and undoubtedly deserves to be published in a journal with some additions.
Author Response
“please see the attachment”

Round 2
Reviewer 1 Report
Comments and Suggestions for Authors
This reviewer appreciates the swift and comprehensive responses from the authors. They have appropriately addressed the previous comments. A few minor points listed below.
The authors changed the recipe calculation of MS medium (L107-L108). I wish the authors to clarify if they used the 1x strength MS or 1.25x strength MS. 1x strength is 4.33g MS salt being dissolved into 1000mL water, and 1.25x strength is 5.41g into 1000ml. In the Arabidopsis research community, 1x and 0.5x strength MS is commonly used. I never know a research case using the 1.25x strength.
In the response letter, “To accurately reflect experimental conditions, the temperature range was described as 25ËšC±2ËšC in our manuscript”. However, the temperature range was not corrected in the revised manuscript (see P114).
Please include the new figure S3 in the supplementary materials.
Some minor typos of the revised manuscript.
L382, ‘mediate’ revised as ‘mediated’.
L416, delete ‘of’.
L436, ‘during’ revised as ‘duration’? Or the sentence is incomplete?
Delete the paragraph from L451 to L463. It was likely accidentally duplicated from the following paragraph.
Reviewer 2 Report
Comments and Suggestions for Authors
Substantial improvements have been made to the manuscript, and it now meets the standards for publication.
Author Response
“Please see the attachment.
